# Exposure to Green, Blue and Historic Environments and Mental Well-Being: A Comparison between Virtual Reality Head-Mounted Display and Flat Screen Exposure

**DOI:** 10.3390/ijerph19159457

**Published:** 2022-08-02

**Authors:** Rebecca Reece, Anna Bornioli, Isabelle Bray, Nigel Newbutt, David Satenstein, Chris Alford

**Affiliations:** 1Centre for Public Health and Wellbeing, University of the West of England, Bristol BS16 1QY, UK; issy.bray@uwe.ac.uk; 2Erasmus Centre for Urban, Port and Transport Economics, Erasmus University Rotterdam, Burgemeester Oudlaan 50, 3062 PA Rotterdam, The Netherlands; bornioli@ese.eur.nl; 3College of Education, School of Teaching and Learning, Institute of Advanced Learning Technologies, University of Florida, Gainesville, FL 32611, USA; nigel.newbutt@coe.ufl.edu; 4Department of Education and Childhood, Faculty of Arts, Creative Industries and Education, University of the West of England, Bristol BS16 1QY, UK; david.satenstein@uwe.ac.uk; 5Psychological Sciences Research Group, University of the West of England, Bristol BS16 1QY, UK; chris.alford@uwe.ac.uk

**Keywords:** restorative environments, well-being, natural and built environments, historic environments, anxiety, EEG, virtual reality, 360-degree video, immersive technology

## Abstract

Improving the mental health of urban residents is a global public health priority. This study builds on existing work that demonstrates the ability of virtual exposure to restorative environments to improve population mental health. It compares the restorative effects of green, blue and historic environments delivered by both flat screen and immersive virtual reality technology, and triangulates data from psychological, physiological and qualitative sources. Results from the subjective measure analyses showed that exposures to all the experimental videos were associated with self-reported reduced anxiety and improved mood, although the historic environment was associated with a smaller reduction of anxiety (*p* < 0.01). These results were supported by the qualitative accounts. For two of the electroencephalography (EEG) frequency bands, higher levels of activity were observed for historic environments. In relation to the mode of delivery, the subjective measures did not suggest any effect, while for the EEG analyses there was evidence of a significant effect of technology across three out of four frequency bands. In conclusion, this study adds to the evidence that the benefits of restorative environments can be delivered through virtual exposure and suggests that virtual reality may provide greater levels of immersion than flat screen viewing.

## 1. Introduction

Exploring the mental well-being outcomes related to exposure to restorative environments is a global priority. This is due to the poor mental health conditions of global citizens, particularly in urban settings. In 2019, almost 1 billion people globally suffered from a mental health condition, such as depression or anxiety [1]. Worryingly, around 20% of children and adolescents globally have a mental health condition [2]. The majority of the population live in urban areas in many parts of the world including the UK [3]. Health can be impacted by the environment in which we live, and it has been well documented that urban environments in particular can have a negative impact on health [4,5], including associations with increased communicable and chronic disease, mental disorders, and suicide [6]. In particular, exposure to traffic in urban environments can be stressful and have a negative impact on health [7]. However, extensive research has shown that restorative environments, such as green (e.g., parks, allotments, and community gardens) and blue spaces (e.g., lakes, rivers, and canals), can have a positive impact on well-being [8,9,10,11,12,13]. Numerous systematic reviews have demonstrated exposure to green spaces and nature to be beneficial for well-being [14,15,16], as well as exposure to blue spaces [17,18]. There is also evidence that virtual exposure to these environments can be beneficial [19,20,21]. The current study assessed the effects of exposure to different environment types (green, blue, and historic) and of different modes of exposure (flat screen (FS) and virtual reality (VR)) on psychological and physiological states on a sample of university students.

There are several theories that can explain the restorative effects of such environments. According to Kaplan and Kaplan’s Attention Restoration Theory (ART) [22], any environment which presents the restorative properties of being away, ‘soft’ fascination, compatibility and extent can also offer restorative benefits. Additionally, Stress Reduction Theory (SRT) posits that being in a natural, unthreatening environment can assist in recovery from stressful situations [23,24]. Attention can be ultimately directed towards natural elements and away from negative emotions, leading to a reduction in stress. Despite the known negative associations between urban living and health outcomes [25,26], recent studies have shown that exposure to certain urban environments can be beneficial for well-being [27,28]. Historic and cultural environments especially have emerged as offering the most restorative potential of urban environments [21,29,30,31,32,33,34,35,36], although the restorative effect of historic settings seems to be smaller than that of natural environments [30]. The restorative power of historic environments can be explained, on one level, by physical features typical of historic environments, such as a high aesthetic value, moderate level of complexity and variety and biophilic design, which all are supportive of restorative perceptions [37,38]. In addition, historic and cultural environments tend to be containers of meaning and trigger place identity and attachment [32,39], which are also known to support restoration [20,40]. It follows that urban environments should be included in research into the well-being benefits of restorative environments, as well as green and blue spaces. Urban environments may not all have green and blue spaces, but may have historic spaces, so it is important to research these as well.

Recent evidence indicates that virtual exposure to natural environments, as well as in-person exposure, can lead to restorative benefits [41,42,43]. This finding opens up several possibilities for public health improvement for populations with limited access to the outdoors (e.g., due to mobility issues or being restricted to confined environments). These populations can include, but are not limited to, prisoners, hospital and nursing home patients, and older adults with mobility issues [44,45,46,47]. Additionally, the recent COVID-19 restrictions, which led to the population being told to not leave their homes, resulted in limited access to outdoor environments. This provides a further motivation to investigate the impact that virtual environments can have on well-being.

There are several ways in which environments can be virtually presented. These can include high-definition TV, 360-degree VR, and computer-generated VR [48]. While 360-degree VR allows the user to view a real, pre-recorded environment in a 360-degree view, computer-generated VR exposes the user to a simulated, artificial environment. Virtual nature environments can create a sense of presence and immersion [49,50] which can influence emotional reactions [51] and the ability of different simulation methods to deliver these benefits has been compared. For instance, by using psychometric measures and assessing heart rate, Brivio et al. [52] compared 360-degree VR and computer-simulated environments as tools to generate relaxation and a sense of presence in participants. Participants viewed two relaxing environments which included elements such as trees, water, flowers, bridges, houses, and natural sounds. They found no differences between the simulation methods used but this study demonstrated the feasibility of using 360-VR environments for passive viewing. Another study compared high-definition TV, 360-VR video and computer-generated VR for viewing an under-water environment [48]. They found that computer-generated VR was more beneficial for mood than 360-VR video or TV and induced a sense of presence and increased nature connectedness. Conversely, when Knaust et al. [53] compared 360-degree VR with a PC monitor when measuring the relaxation effects of a beach scene, they found no differences between VR and PC for psychophysiological measures (skin conductance and heart rate); however, VR was rated as more relaxing than PC when assessing self-reported relaxation. In conclusion, it is still unclear in the literature how experiences of viewing restorative green and blue environments through VR compare to simple viewing on an FS PC monitor. It is important to understand these potential differences so that the most beneficial exposure method can be adopted in future research, as well as in practice, for populations with limited access to real environments.

Taken together, there is growing evidence that virtual nature environments can support psychological well-being and restoration. Turning to the well-being potential of virtual exposure to urban environments, there is some evidence that FS exposure can be beneficial [21]; nevertheless, there is a substantial lack of evidence on the effects of VR exposure to positive urban environments, and we aimed to address this gap by including a historic environment in our study.

A key suggestion for future research made by Knaust et al. [53] is to explore psychophysiological responses using electroencephalography (EEG), given the limited evidence available in relation to physiological outcomes. EEG has been used in combination with VR in several contexts, e.g., in relation to neurorehabilitation [54,55] and mindfulness training [56,57]; however, its use with virtual restorative environments is limited. Several studies have measured physiological and psychological well-being [52,53]. EEG measurement provides an objective measure of brain cortical activity and has been proposed as an ideal measure of relaxation [58]. EEG changes have reflected improved subjective mood states when viewing green environments [59,60], though results may vary and change with presentation method [61,62]. Few have gone beyond this to triangulate these measures with qualitative findings. One study did triangulate EEG with interview data [63], although studies like this are generally lacking. This is important as triangulation can provide a more in-depth and reliable understanding of how exposure to virtual restorative environments can impact well-being.

The current study aimed to assess whether exposure to a potentially restorative environment (green, blue, and historic), presented either through VR or FS video, was associated with mental well-being benefits in terms of reduced self-reported state anxiety (measured by the STAI) and stress (measured by the UWIST MACL), as well as physiological measures of improved mood and relaxation (EEG), following exposure to a stressor environment (traffic). This study also aimed to triangulate results from subjective (psychological) and objective (physiological) measures of well-being, and responses to qualitative open-ended questions. To address these aims, the following research questions were formulated:RQ1: Is virtual exposure to green, blue, and historic environments associated with reduced self-reported anxiety and stress, and with physiological measures of relaxation, compared with virtual exposure to a traffic environment?RQ2: Are there differences in effectiveness between VR and FS exposure to green, blue, and historic environments?

## 2. Materials and Methods

### 2.1. Participants

This study was conducted at a university in England, United Kingdom. Participants were a convenience sample who responded to the study advertisement through the university psychology participant pool, flyer advertisement on course noticeboards and in emails, and through social media advertisement. Recruitment took place between June and December 2021. Participants eligible to take part were healthy volunteers with self-reported normal hearing and normal or corrected-to-normal vision. Participants were excluded if they suffered from a skin allergy or hypersensitive skin, had experienced an epileptic seizure within the past 12 months, had current or previous hypertension, anxiety, psychiatric, neurological disorders, or current illness (e.g., COVID-19, and influenza), and/or were taking prescribed medication for brain or psychiatric conditions (e.g., anxiety, depression, or epilepsy).

In total, 31 participants took part in the study, of which 22 were female (71%). Participants ranged in age from 18 to 57 years (M = 24.32, SD = 8.73). All participants were students at the university, with 27 currently enrolled on an undergraduate psychology program (87%). Half of participants indicated White-British ethnicity (52%), and 26% indicated White-Other ethnicity. Participants were randomly assigned to the VR or FS condition (Table 1). Data from all 31 participants were included in the analysis of subjective (psychological) measures. During EEG analysis, data relating to three participants was removed due to high impedances (resistance to the flow of alternating current) or errors in recording; therefore, 28 participants were included in the analysis of objective data.

### 2.2. Design

A between-subjects design was employed and involved participants watching three environment videos (green, blue, and historic) and three stressor videos (traffic); see Figure 1. Participants were randomly assigned to watch the environment videos on a FS computer monitor or using a VR head-mounted display (HMD). Regardless of condition, stressor videos were 30 s in duration and viewed on an FS computer monitor before each environment video, with the aim of bringing participants back to a more negative mood. Subjective mood was measured after each stressor and environment video. EEG was recorded continuously throughout the experiment. Qualitative comparative assessment was made at the end after experiencing all 3 environments.

The order of presenting restorative environment videos (green, blue, and historic) was also randomized. Twelve participants viewed the green environment first, ten participants viewed the historic environment first, and nine participants viewed the blue environment first.

### 2.3. Exposure to Restorative and Stressor Environments

Three videos were filmed for the experiment. These included a green environment, blue environment, and historic environment (Figure 2). The green environment showed an open field with vast views of the Somerset countryside. Grass, bushes, deciduous trees, and fields with sheep can be seen in the distance. The weather is clear and sunny, and the sound of the wind can be heard. The blue environment depicted a large lake (Chew Valley Lake, UK) with trees and vegetation visible on the horizon. Ripples in the water are visible and the sound of the water can be heard. The weather is cloudy with some sunshine. Behind the camera (viewable by turning around 180 degrees with HMD), there are trees, bushes and grass, as well as a public footpath following the edge of the lake. The historic environment depicted a built historic street dating from the 14th century (Vicars’ Close, Wells, UK) with cobbled-stone pavements and views of a cathedral behind the camera. The street comprises of several Grade I listed buildings. A bird is seen flying around rooftops. The weather is sunny with some cloud and the bushes can be seen moving in the wind. Near the end of the video, some people are visible (no people are visible in the blue and green environments). Each environment video lasted two minutes. Participants viewed all environment videos either sat 46 cm away from an FS Hewlett-Packard computer monitor (57 cm × 34 cm) or using a VR HMD (Oculus Go HMD). A 30-s video of heavy traffic on an FS computer monitor preceded each of the environment videos and acted as a stressor, following similar procedures to earlier research (e.g., Van den Berg et al. [64]).

### 2.4. Subjective Measures

The shortened version of the State-Trait Anxiety Inventory (STAI) [65] was used to measure trait anxiety, reflecting general propensity to experience anxiety at the beginning of the experiment, and state anxiety, reflecting the level of anxiety currently experienced after viewing each environment. The trait anxiety scale includes six statements; ‘I am calm, cool and collected’, ‘I worry too much over something that really doesn’t matter’, ‘I feel secure’, ‘I get in a state of tension or turmoil as I think over my recent concerns and interests’, ‘I feel nervous and restless’, and ‘I make decisions easily’. Participants are required to answer on a 4-point scale how they feel (almost never, sometimes, often, almost always). The state anxiety scale included includes six statements; ‘I feel calm’, ‘I am tense’, ‘I feel at ease’, ‘I feel nervous’, ‘I feel relaxed’, and ‘I am worried’. For this scale, participants are required to answer on a 4-point scale of how they feel (not at all, somewhat, moderately so, very much so). STAI showed alpha’s reliability coefficients, per environmental condition, ranging from 0.73 to 0.94.

Non-clinical mood was measured using the shortened version of the University of Wales Institute of Technology Mood Adjective Check List (UWIST MACL), which has shown to have good validity and internal reliability [66]. This scale includes four items (relaxed, nervous, happy, and sad). The output from this measure includes a score for hedonic tone, stress, and energy. For this study, stress was calculated by combining scores for nervous and reversed scores for relaxed [66]. Stress showed alpha’s reliability coefficients, per environmental condition, ranging from 0.61 to 0.76.

Both scales were completed by participants on Qualtrics (Qualtrics, Provo, UT, USA, 2022) after viewing every video.

### 2.5. Objective Measure

EEG is an objective measure of brain activity and can be used to assess levels of physiological stress and relaxation [67,68], but has yet to be used to assess relaxation in VR environments [58] and was therefore continuously recorded during this experiment. Previous research has demonstrated the utility of EEG as a psychophysiological measure including changes in affect with increased left hemisphere activity reflecting more positive mood in some studies, though not all [69,70], and pleasant emotion with increased midline theta [71]. Increased stress is associated with increased beta but reduced alpha frequencies [72], and with green exposure reducing beta frequency including when associated with traffic [73,74]. EEG changes in alpha, beta and gamma frequencies have been linked to changes in relaxation [59] and EEG has been proposed as an ideal measure of relaxation ‘R-state’ by Zhang et al. [58], with increases in relaxation reflecting restorative environments, such as increased alpha in response to passive viewing of rural images [73] and increased frontal and occipital alpha when viewing green rather than urban spaces [75]. Similarly, relaxed attentional states, as opposed to loss of attention and vigilance, have been associated with increased alpha and theta [61]. Differences between ‘real’ and simulated environments have also produced changes in EEG [41]. Based on the literature, EEG electrode grouping for this study included frontal contrasted with occipital areas, as well as lateral contrasted with medial electrodes and left with right hemisphere activity. The frequency bands assessed included theta, alpha, beta and gamma frequency ranges.

Measurements were recorded using a non-invasive cap with 32 electrodes and electrode gel was used to ensure low impedances. The electrode channels were connected to a QuickAmp amplifier. BrainVision Recorder 2 was used to record the data (BrainVision Recorder, Version 1.24.0101, Brain Products GmbH, Gliching, Germany) and a sampling rate of 1000 Hz and notch filter of 50 Hz were adopted. Markers were placed on the recording when video stimuli started and finished.

#### 2.5.1. Pre-Processing

EEG data were analyzed using BrainVision Analyzer 2 (BrainVision Analyzer, Version 2.2.0, Brain Products GmbH, Gilching, Germany). The dataset was bandpass filtered using a FIR filter with low cut-off 0.1 Hz and high cut-off 100 Hz. Pooling was conducted to create new averaged channels for electrodes with high impedances. The data were segmented into sections for traffic and restorative environments and were then further segmented into 10 s segments. Following segmentation, the data were examined for physical artefacts (e.g., muscle movements) using semi-automatic inspection and segments were removed if excessive artefacts were present. Ocular correction with independent component analysis was implemented to remove eye blinks. The data were re-referenced to the common average reference and sampling rate was adjusted to 512 Hz before segmentation was conducted to divide the data into two second segments.

#### 2.5.2. Spectral Analysis

Using the Fast Fourier Transform, the data were transformed from time to frequency domain. Spectral power (µV²) was exported for theta (4–8 Hz), alpha (8–13 Hz), beta (13–30 Hz), and gamma (30–80 Hz) frequencies. Overall frontal activity was averaged over electrodes Fp1, Fp2, Fz, F3, F4, F7, and F8. Medial frontal activity was averaged over electrodes F3 and F4. Lateral frontal activity was averaged over electrodes F7 and F8. Left frontal activity was averaged over electrodes F3 and F7, and right frontal activity was averaged over electrodes F4 and F8. Overall occipital activity was averaged over electrodes Oz, O1, and O2.

### 2.6. Additional Measures

Since we were also interested in the effects of personal preference for different environments and the role of immersion, we asked some additional questions to measure these concepts. A questionnaire completed at the end of the experiment, on Qualtrics, asked participants to state their preference for urban and natural environments using a 5-point scale (1 = ‘strong preference for natural environments’, 5 = ‘strong preference for urban environments’). The questionnaire also included, “To what extent do you immerse yourself within environments just to appreciate and observe them”, and participants could choose ‘very often’, ‘often’, ‘not sure’, ‘sometimes’, or ‘never’. After watching each environment, participants were also asked whether they were familiar with the environment. Additionally, socio-demographic data were collected (age of participant, gender, and ethnicity).

### 2.7. Qualitative Outcomes

Seven optional open-ended questions were also included in the final questionnaire. These questions asked; ‘How was your experience of watching the videos?’, ‘Is there anything in particular that attracted your attention in the videos?’, ‘To what extent did you find the experience realistic?’, ‘To what extent did you feel immersed in the environment?’, ‘How relaxing/engaging was it watching the videos? If it was relaxing or engaging, why?’, ‘Do you have any comments on the quality of the videos? Did you identify any technical issues?’, and ‘Do you have any additional comments?’.

### 2.8. Procedure

Upon arrival in the lab, participants read a participant information sheet and signed a consent form before taking part in the study. Next, EEG electrodes were set up and attached ready for recording. Before beginning, participants completed the shortened STAI questionnaire. Participants then completed the experiment (see Figure 1). Afterwards, participants were thanked and debriefed. Participants were rewarded for their time with the choice of course credits or a voucher. The total duration of the experiment was approximately 1.5 h. Data were analyzed using IBM SPSS Version 26 (IBM Corp., Armonk, NY, USA) [76] and Stata Version 16 (StataCorp, College Station, TX, USA) [77].

### 2.9. Ethics

Ethical approval was granted from the Faculty Research Ethics Committee (FREC) from the University and was in accordance with the Declaration of Helsinki. To avoid cybersickness when using the VR HMD, participants were offered breaks and water in between videos. A full risk assessment was completed for the study, as well as a COVID-19 risk assessment, and were approved before commencing.

## 3. Results

### 3.1. Subjective Measures

#### 3.1.1. Descriptive Statistics

Table 2 displays descriptive statistics for the subjective outcomes measured after exposure to each environment. Compared to the traffic video, state trait anxiety and stress were lower in all of the experimental conditions.

Participants reported their preference for natural and urban environments. Overall, most participants expressed either a strong or slight preference for natural environments (87%). Three participants had a slight preference for urban environments (10%), and one participant had no preference (3%). Participants also reported on how often they immersed themselves in environments just to appreciate them. Most participants responded ‘sometimes’ (58%), ten responded ‘often’ (32%), and three responded ‘very often’ (10%). Participants were not very familiar with the restorative environments that they viewed (Figure 3).

#### 3.1.2. Linear Mixed-Effects Models

Mixed-effects regression models were fitted using Stata’s code mixed. These predicted differences in subjective measure scores between each experimental condition and the preceding traffic video based on arm of trial (method of virtual exposure) and controlling for age, gender, and order of experimental videos as fixed effects. The State Trait Anxiety Index model also controlled for baseline trait anxiety as a fixed effect. Table 3 shows changes in subjective state after exposure to the environments with lower scores reflecting reductions in perceived anxiety or stress. There were no statistically significant differences in changes pre–post exposure for anxiety and stress scores between the three environmental conditions, except for the historic environment which was associated with a smaller reduction of state anxiety compared with the blue environment. VR exposure was not statistically different from the FS exposure.

### 3.2. Objective Measures

Statistical analyses were conducted to compare brain activity between environments (green, blue, and historic) and between presentation conditions (FS vs. VR). Four frequency bands were investigated and compared (theta, alpha, beta, gamma), across several locations in the brain (overall frontal, overall occipital, medial frontal, lateral frontal, left frontal, and right frontal). Separate Mixed ANOVAs were performed which included a within-subjects factor: environment (green, blue, and historic), and between subjects factor: technology (FS or VR). Mean activity was averaged across traffic videos to give an overall value for the stressor environment. All analyses used two-tailed significance levels (*p* < 0.05).

#### 3.2.1. Theta

Two-way mixed ANOVAs were performed on the level of theta activity in frontal and occipital brain regions when participants viewed three environments either using VR or FS (technology). The main effect of the within-subjects factor ‘environment’ was statistically significant for theta activity in the overall frontal (F(2,52) = 5.28, *p* = 0.008, η2p = 0.17), occipital (F(2,52) = 6.12, *p* = 0.004, η2p = 0.19) and right frontal (F(2,52) = 4.56, *p* = 0.015, η2p = 0.15) regions of the brain, with the highest levels of theta for the historic environment (Figure 4).

The main effect of the between-subjects factor ‘technology’ was statistically significant for theta activity in the overall frontal (F(1,26) = 18.19, *p* = < 0.005, η2p = 0.041), medial frontal (F(1,26) = 4.33, *p* = 0.047, η2p = 0.14), lateral frontal (F(1,26) = 9.29, *p* = 0.005, η2p = 0.26), left frontal (F(1,26) = 11.41, *p* = 0.002, η2p = 0.30), and right frontal (F(1,26) = 11.17, *p* = 0.003, η2p = 0.03) regions of the brain. These regions of the brain highlighted increased theta activity in the VR condition compared to FS, for all environments. There was also a statistically significant two-way interaction: environment by technology: (F(2,52) = 3.35, *p* = 0.043, η2p = 0.11) for theta activity in overall frontal regions.

Overall theta activity whilst viewing the traffic videos was higher in all regions of the brain than when viewing environments (green, blue, and historic) on a FS, but lower than when viewing environments using VR (see Appendix A). One exception was that participants had higher theta activity in the occipital region of the brain when viewing the traffic video compared to viewing the blue environment, regardless of technology condition.

#### 3.2.2. Alpha

The main effect of the within-subjects factor environment was statistically significant for alpha activity in the medial frontal region of the brain (F(2,52) = 3.99, *p* = 0.024, η2p = 0.13). The highest mean alpha activity in the medial frontal region was for participants viewing the historic environment using VR (M = 0.85), whereas the lowest mean alpha activity was for participants viewing the green environment on a FS (M = 0.61). The average alpha activity in the medial frontal region when viewing traffic was in between (M = 0.72) (see Appendix A).

Additionally, there was a significant main effect of the between-subjects factor technology in the frontal regions (overall average) (F(1,26) = 5.44, *p* = 0.028, η2p = 0.17), and the right frontal region of the brain (F(1,26) = 5.07, *p* = 0.033, η2p = 0.16). There was higher mean alpha activity for participants viewing all environments using VR, compared to participants viewing environments on a FS. For frontal regions (overall average), alpha activity was higher when viewing environments in VR compared to traffic, but higher activity was found when viewing traffic compared to FS. For the right frontal region of the brain, there was higher alpha activity when viewing traffic than when viewing green, blue and historic environments on FS. Viewing the historic environment with VR was the only condition when right frontal alpha activity was higher than when viewing traffic.

Two-way mixed ANOVAs were also performed to assess differences between the left and right cortical hemispheres in frontal alpha activity. There was a statistically significant main effect of the within-subjects factor hemisphere for participants viewing the blue environment (F(1,26) = 10.09, *p* = 0.004, η2p = 0.28). For the green environment, there was also a statistically significant effect of hemisphere (F(1,26) = 5.46, *p* = 0.027, η2p = 0.17) (Figure 5).

For both the green and blue environments, more alpha activity was recorded in the left frontal region of the brain compared to right frontal, for both FS and VR conditions. Additionally, when participants viewed traffic, there was less alpha activity in left frontal regions than right frontal regions. There was no statistically significant effect for hemisphere for the historic environment and no statistically significant effect of technology or interaction effects were found for any environment.

#### 3.2.3. Beta

No statistically significant main effects or interaction effects were found when assessing beta activity across frontal and occipital regions of the brain. As with theta and alpha, there was overall increased activity for the VR condition compared to FS, but there was no significant effect of technology (see Appendix A). However, interestingly, beta activity was reduced during the VR condition, compared to FS, when participants viewed the blue environment. Levels of beta when participants viewed traffic were lower than when viewing the blue environment, regardless of exposure type (VR or FS).

#### 3.2.4. Gamma

Two-way mixed ANOVAs were also conducted on the level of gamma activity. There was a statistically significant main effect of technology in the frontal regions (overall average) (F(1,26) = 4.97, *p* = 0.035, η2p = 0.16) and medial frontal region of the brain (F(1,26) = 6.07, *p* = 0.021, η2p = 0.19). Higher gamma activity was observed in these areas of the brain when participants viewed all environments using VR compared to participants viewing on a FS (See Appendix A).

Additionally, for frontal regions (overall average), higher gamma activity was recorded for participants viewing the traffic video compared to viewing all FS environments, but lower compared to VR experiences. This was similar for medial frontal activity, except gamma activity whilst watching traffic was lower than viewing a green environment (regardless of technology condition). No statistically significant results were found for the main effect of environment or an interaction effect: environment by technology.

### 3.3. Qualitative Outcomes

In analyzing these open-ended questions, we created four categories from the seven questions/prompts (detailed above in the Methods section). These were: (1) experience and immersion (questions 1,3,4); (2) attention in the experience (question 2); (3) relaxing and/or engaging (question 5); and (4) technical/quality related (questions 6,7). The data below draw upon direct quotes. We used a system to identify the condition they were assigned (VR or FS), followed by their gender (M or F) and finally their age. For example, VR-M-45, refers to a male, aged 45 in the VR condition.

Considering the first category (experience and immersion), participants from across both conditions (VR and FS) reported the experience of watching the environments (green, blue, and historic) to be “enjoyable and interesting, some of them were very calming” (FS-F-19) and “the ones of nature were very relaxing in comparison to the ones of traffic” (VR-F-18). The content of the scenes also generated positive comments, with one participant in the VR condition stating: “Virtual reality videos were very pleasant, the nice weather in all of them helped me feel at ease” (VR-F-19). Comments related to the relaxing nature of the 360 videos were echoed across both the FS and VR conditions, with participants reflecting: “I really enjoyed both the countryside and water based environments—I felt significantly more relaxed during those” (FS-F-20), “The experience was good, found the nature videos a lot more relaxing than the traffic ones” (FS-F-21), “Interesting and generally calming” (VR-M-20), and “I found the videos on the VR headset very relaxing, and in a much calmer environment than watching the busy roads” (VR-F-20).

In relation to feeling immersed, the VR condition created far more detailed responses. The FS environment elicited 278 words in total comments with the VR condition eliciting 612 words of comments. The FS condition participants reported the realism and immersion to be: “somewhat” (FS-F-35), “not really” (FS-F-29), “fairly” (FS-F-20), and “slightly” (FS-F-22). Additionally, one commented: “[it was] pretty realistic for a lab experiment” (FS-F-21). The FS condition was not without more positive comments related to immersion and realism, with several comments suggesting: “Well I suppose it is realistic enough” (FS-F-49), “very, there was a lot to look at and take in within all the videos/scenery. With the sea scene especially as you could just watch the waves rather than looking around at everything else” (FS-F-19), and “very much with the residential area and the video with the water” (FS-F-35). Similarly, the VR condition also generated positive comments suggesting that participants felt: “Quite immersed, the sound definitely made it more immersive and the response to movement, I felt like I could’ve talked to the people like it was a live recording instead of recorded at an earlier date” (VR-M-22), “moderately so” (VR-F-20), “not sure” (VR-M-28), and “I didn’t feel very immersed in the environment. I think maybe I was too aware that what I was seeing wasn’t real and that I couldn’t see my actual surroundings was perhaps a little disconcerting” (VR-F-28). However, these were the only negative comments related to the VR exposure, with far more positive comments overall, such as: “[…] the VR videos they definitely had my whole attention” (VR-F-20), “VR felt much more realistic” (VR-F-20), and “very realistic, the sound really made me feel like I was there, was feeling very similar to being there” (VR-M-22).

The second category was related to attention in the experience. The FS experience generated several comments about the stressor videos (i.e., cars) while the VR experience generated none. The FS comments related to the scenery (green, blue, and historic) included: “The video of the sea and the final video of the green area did remind me of where I am originally from (Sussex) […] the green especially did remind me of the south downs” (FS-F-19), “I liked the water-front as well as seeing the old man taking a walk on the stone road” (FS-F-18). The VR condition generated more detailed and specific comments related to attention. Two specific examples were: “I found whenever there were people in the videos my attention was drawn to them” (VR-F-28) and “the horizon, was a focal point for me” (VR-M-28).

For the third category, relaxing and/or engaging, comments from participants in the FS condition suggested that they were mixed: “the video with the massive green area was not relaxing, I felt alone” (FS-F-35) and “it was quite relaxing, felt like it was real because I usually go to the forest just to refresh when I’m happy or sad” (FS-F-22). Additionally, one participant suggested the lack of elements like wind or smells limited their relaxing in the FS environment(s), saying: “not engaging as I cannot feel the wind or smell the animals in the country […] relaxing a little as it allows you to reflect when looking onto the countryside” (FS-F-29). On the other hand, some FS participants felt their experiences were relaxing stating: “The scenery videos were very relaxing, although at first, I thought I should be looking for something in particular or there may be something that suddenly happens or changes […] this meant the final scenery ones were more relaxing than the first” (FS-F-18) and “very relaxing to watch the natural environments but less so the urban ones” (FS-F-20). Some felt the FS environments with less going on were linked to them feeling relaxed: “Relaxing - rural areas more chilled less to observe” (FS-F-30), with some feeling less going on was linked to lack of engagement, stating: “The countryside and lake ones were very relaxing. Not as engaging though” (FS-F-20). Comparing this to the VR condition, all participants revealed they were/felt relaxed, and referred to several physical elements of the VR-360 experiences. Comments included: “Very relaxing as you could imagine being there, listening to the wind blowing and the reflection of the sun going behind the clouds” (VR-F-57); “it was relaxing as it gave me a time to relax and be creative and think fun thoughts like touching the grass and cobbles and imagining being on a day out adventure on my own in the sun” (VR-M-22); “very I liked being able to look around and could hear the wind, water” (VR-F-20); “very relaxing due to the nice environments, with very few other people around” (VR-M-21); and “the naturalistic environments were particularly relaxing, they had a calm centered quality that I felt allowed you to let go of thoughts more easily” (VR-M-28).

Finally, the fourth category, technical/quality, raised issues of the fidelity and quality of the VR condition, with several comments about the material/videos being “ever so slightly blurry” (VR-F-20); “the VR footage was a little grainy” (VR-F-28); and “the videos on the VR could have better quality” (VR-M-22). The FS condition yielded several comments related to the video(s) being “good” and that “they were all high quality and realistic and I had no technical issues” (FS-F-19), but the participants also suggested there was room for improvements, stating: “The quality was a little poor in some” (FS-F-20) and that the “nature videos were a little bit blurry” (FS-F-21).

## 4. Discussion

This study investigated responses to exposure to different environments (green, blue, and historic) compared with a stressor video, with different tools (FS and VR). Measurements included psychological, physiological, and qualitative outcomes. Two research questions were posed; (1) Is virtual exposure to green, blue, and historic environments associated with reduced self-reported anxiety and stress, and with physiological (EEG) measures of relaxation, compared with virtual exposure to a traffic environment? (2) Are there differences in effectiveness between VR and FS exposure to green, blue, and historic environments?

Results from the subjective measure analyses showed that exposures to all the videos of restorative environments were associated with reduced self-reported anxiety and stress. There was little to differentiate the effectiveness of these three environments, except some evidence of a smaller reduction in anxiety following the historic environment. These results were supported by the qualitative accounts, which suggested that the restorative environments were relaxing and calming, but the historic environment perhaps less so than the green and blue environments. Findings for the physiological (EEG) measurements were less conclusive. For two of the frequency bands (theta and alpha), significant differences were found between the experimental environments (green/blue/historic), with activity being the highest for historic environments. For the other two channels, there was no evidence of any differences between the environments. In relation to the role of mode of delivery, the quantitative subjective measures did not suggest any effect, while for the EEG analyses activity was generally higher for VR than FS, and there was evidence of a significant difference across three of the four frequency bands. Whilst the subjective ratings of stress (UWIST MACL) and anxiety (STAI-S) did not result in significant differences between presentation formats (FS contrasted with VR), these EEG changes were aligned with some qualitative descriptors reflecting greater experiences of immersion, presence and realism for the VR presentations resulting in greater levels of cortical brain activity (EEG power). Increased alpha and theta with VR exposure may also reflect greater ‘relaxed attentional states’ as proposed by Lomas et al. [75].

The self-reported psychological responses, which improved in the green, blue, and historic conditions compared with the traffic video, confirmed previous research that exposure to green [8], blue [43] and historic [32] environments can offer mental well-being benefits, and partially answered research question 1. The blue environment was associated with a larger reduction of state anxiety compared to the historic condition, and was not statistically different from the green condition, indicating that reduction of anxiety is maximized in natural environments, as opposed to historic built ones [30]. Conversely, where the EEG results suggested there were differences in restorative effect between the environments (theta and alpha), it was the historic environment that appeared to have the largest effect, an apparent contradiction in the findings for research question 1. It is possible that historic settings can support other forms of mental well-being benefits, such as triggering soft attention stimulation or cognitive engagement [39] again reflecting relaxed yet attentional states [75], rather than stress reduction and relaxation which are properties of natural environments, and that these were detected by the EEG measurements.

An increase in alpha activity can reflect decreased cortical activation and be likened to relaxation with alpha reflecting a resting frequency for cortical activity [78] and increased cortical activity may reflect greater engagement. Our results are consistent with other studies which have found green spaces to be associated with increased levels of meditation [60] and provides partial support for increased alpha activity indicating increased relaxation and reduced visual processing [72,79,80]. Our study also showed significant increased alpha activity in the left frontal region when participants viewed the green and blue environment, regardless of exposure type (FS or VR). This may be explained by higher levels of alpha representing increased cortical activity, as opposed to inactivity, and fit with positive emotions associated with being processed in the left hemisphere, and less positive emotions associated with the right hemisphere [81]. Although significant increases in subjective hedonic tone/positive mood state were not found in the current study, limiting comparative assessments with EEG, alpha levels overall in the traffic condition were higher than for FS natural environments overall. Earlier research has shown differences between viewing real environments and video presentation [61]. There was further support for the significant increase in medial theta with VR presentation, perhaps reflecting more positive mood overall [70], linked to increased engagement and immersion in our study. Previous studies have shown significant changes in beta activity when participants view blue spaces [61] and green spaces, compared to urban settings [72]. Further, beta reductions were reported for green environments [72] and reflected reduced traffic induced stress when viewing green environments [73]. There were no significant changes in beta activity in the present study which therefore differs from previous literature and does not provide support for reductions in beta activity reflecting reductions in stress. Nevertheless, some other studies have also found no significant differences in activity between environments which varied in grey, green, and blue space [41,62] and may reflect the developing status of these relatively new research applications. In regard to historic environments, to the best of our knowledge this is the first study which assessed EEG outcomes in these settings, hence the findings that historic environments can trigger increased alpha and theta, reflecting improved cortical engagement and relaxation, is novel and warrants further research.

While the psychological measures did not detect any difference according to mode of delivery, limiting comparisons, the physiological measures (EEG) show a significant effect of technology. Increased brain activity was recorded across multiple frequencies when viewing all environments through VR, compared to FS exposure and baseline stressor (FS traffic). This effect of technology is consistent across theta, alpha and gamma frequencies, lending weight to this finding in response to research question 2. In previous studies, meditation and relaxation have been associated with increased occipital gamma activity [74,82]. However, this is not consistent with the findings from the present study which showed increases in frontal gamma activity when participants viewed the environments through VR compared to FS, but no significant difference for occipital gamma. However, as mentioned, the significant increases in both alpha and theta frequencies, reflecting a state of relaxed alertness, are supported by previous research [75].

We suggest that the different findings for psychological and physiological measures could potentially be explained by considering evolution and genetics. As animals, humans have an innate tendency to be in nature and connect with nature, which is explained by the biophilia hypothesis [83]. This may be why the enhanced immersiveness provided by VR had an effect that could be detected physiologically through brain activity changes which may be subconscious, but not through self-reported measures, which measure conscious perceived psychological state. Further, genetics may play a role and there may be genetic differences among people’s connection with nature (see, for example, the twin study by Chang et al. [84]). At the same time, it is possible that historical sites in particular can engender a sense of awe and wonder [21,85], which may explain why this study uncovered an effect of the historic environment for EEG measures. Knowing that most of the population live in urbanized areas [86], and that the amount of urbanization in our hometowns can influence our preferences for environments [84], this is a finding which can inform policies for healthy cities. Although beyond the scope of this paper, further research could investigate the association between individual objective brain responses to different environments and subjective environmental preferences, dwelling location and other person-specific factors.

In relation to research question 2, no significant effects were found for technology (VR vs. FS) in terms of self-reported anxiety. This could be explained by participants potentially having some anxiety with using VR headset, as novel technology, which may have cancelled out any effects of the more realistic exposure. One participant stated, “...I couldn’t see my actual surroundings was perhaps a little disconcerting” (VR-F-28). This explanation can also be viewed in a positive light, in that VR did not increase anxiety in participants, meaning VR did not create a negative experience. Further, it was reported that no participants suffered symptoms of cybersickness during the study, which is a surprising finding knowing the prevalence of cybersickness effects of VR [87]. However, some research has stated that cybersickness happens after 10–15 min of immersion [88], which may be why the two minutes of exposure at a time in this study did not have an effect.

Additionally, in relation to research question 2, this study has shown through physiological measures and qualitative responses that there are differences between exposure type for viewing green, blue, and historic environments. Taken together, the qualitative data (captured through post-experience surveys) revealed that participants preferred the VR experience for promoting feelings of relaxation, in addition to a sense of realism and ‘being there’ (ecological validity). This can lead to increased perceived presence [89]. The FS condition led to more comments that suggested lower levels of realism or immersion, but in terms of attracting attention, the FS and VR conditions elicited similar responses and replies. Qualitative responses provide most support for there being a difference in technology (VR vs. FS), which was consistent with our physiological findings. Qualitative responses can be used to further explain the differences between technologies. VR provides a space to relax but also be attentive (e.g., focus on a particular aspect of that environment, such as trees and wind, etc.). This is consistent with the EEG findings in relation to the increase in alpha and theta activity which can reflect ‘relaxed attentiveness’ [75]. Theta power increases have also been explained as underlying focused attention and ‘getting in the zone’, and alpha increases reflecting focused attention, with local inhibition of unwanted cognitive activity such as distractors, in successful sports performance [70]. This would indicate for the present study that participants were in a state of focused attention whilst viewing environments through VR compared to FS. This could be beneficial for populations such as people with autism by providing a means to relax and escape into a virtual space [90], whilst also being able to focus and direct attention.

Another interesting finding was that participants in the VR condition generally gave more qualitative feedback compared to those in the FS condition. Furthermore, participants in the VR condition only talked about the VR environments in their qualitative responses and not about the traffic videos, whereas the participants in the FS condition also talked about the traffic videos. It is also important to recognize what VR can take away as well as what it can provide (e.g., shutting out distractions in the world around us). This could be an important attribute for exposure to an environment, to maximize the restorative potential.

This study has provided a novel approach by triangulating psychological, physiological, and qualitative outcomes. Firstly, this study highlights the importance of, and differences between, the choice of measures used in research to investigate the well-being benefits of different environments. We found agreement between the psychological measures and qualitative perspectives, although the physiological measures (EEG) did not fully mirror the quantitative psychological results. Secondly, a comparison has been made in this study between FS and VR as exposure types for viewing environments. This investigation also served as an exploratory study for the use of VR as a method of exposure to environments when assessing their well-being potential. Clear differences have been found in EEG recordings between the exposure types, supporting the use of VR as an immersive exposure option in future studies. Thirdly, our results provide some evidence for the positive impact of exposure to historic environments which should be investigated further as it has particular potential for improving the mental well-being of urban populations.

This was a randomized experiment with good balance between the two arms of the trial. However, the sample size in this exploratory study was small though reflecting typical EEG studies, and so the results should be interpreted with caution. The quality of the videos shown was commented on by participants viewing on FS, as well as through VR. It was highlighted by some that videos were blurry and so the quality could have been improved. The Oculus Go headset was chosen for its good usability; however, future research should consider using better quality VR technology and filming equipment for recording videos of environments. Overall, the results suggest that VR is a worthwhile technology, and it warrants further investigation in future studies. Second, there was an imbalance in the number of males and females in the conditions of the experiment. There were more males in the VR condition (*n* = 6) compared to the FS condition (*n* = 3) and more females in the FS condition (*n* = 13) compared to VR (*n* = 9). Furthermore, due to convenience sampling as well as recruitment through the university’s psychology participant pool, which is predominantly made up of female students, the majority of our participants were female. This may have impacted the results as previous research has found that female participants were more excited about using new technology (such as VR), and this technology triggered more positive emotions than with males [91]. Future research should focus on including all sexes and examining differences in relation to the well-being benefits of exposure to virtual environments. Third, the novelty of using VR for the first time may have influenced the experience and results of the participants in the VR condition. Future research can counterbalance for this or only use either first time users or experienced users of VR, or by adding an initial adaptation trial where data were not recorded, but participants can adapt to the novel technology and laboratory protocol. Additionally, participants were seated whilst watching all environments, and this may not have been as immersive as if participants were standing up and physically turning around in the virtual environment.

It should be noted that research into the effects of exposure to different environments using EEG is still in the early stages and these results should be taken as preliminary. There are several competing models that can explain the EEG results and more research is needed to interpret them. Additionally, frequencies coming from the VR headset in proximity to the EEG electrodes could be an artefact and may explain the consistent increase in activity recorded across some frequencies. Some previous research has examined the impact of head mounted displays (VR) technology on concurrent EEG recording but found that whilst there was some electromagnetic interference at higher frequencies, the lower frequencies investigated in the current study (theta 4–8 Hz, alpha 8–13 Hz, beta 13–30 Hz, gamma 30–80 Hz) were largely unaffected, though line noise (50 Hz) could be a factor but was effectively filtered out in our analysis, and a 90 Hz refresh rate for one headset impacted higher frequencies [31]. Future studies might usefully include an additional baseline EEG comparison with and without the VR headset being active in order to isolate and quantify the effects of the VR technology alone on EEG frequencies.

## 5. Conclusions

In conclusion, the psychological and qualitative analyses provided evidence to support the beneficial effects of virtual exposure to natural and built environments on anxiety. There was no apparent difference between the green and blue environments, and some limited evidence that they may provide greater restoration than the historic environment. However, further studies with a wider range of environments and larger sample sizes are needed, also acknowledging the role of individual preferences for particular environments. We note that the novelty of the VR technology may counteract any potential relaxing effects but was found to be promising as a method of presenting potentially restorative environments. Physiological (EEG) findings are harder to interpret but tended to suggest higher levels of brain activity when using VR compared to FS viewing, and in historic environments compared to natural ones. These findings possibly indicate greater relaxed engagement with the environments with the VR technology, and therefore greater immersion. This lends weight to the potential for developing VR technologies to be used to research the effects of exposure on a wide range of different environments, and to deliver these benefits to subgroups of the population who are not able to access such environments (such as prisoners, and those with impaired mobility).

## Figures and Tables

**Figure 1 ijerph-19-09457-f001:**
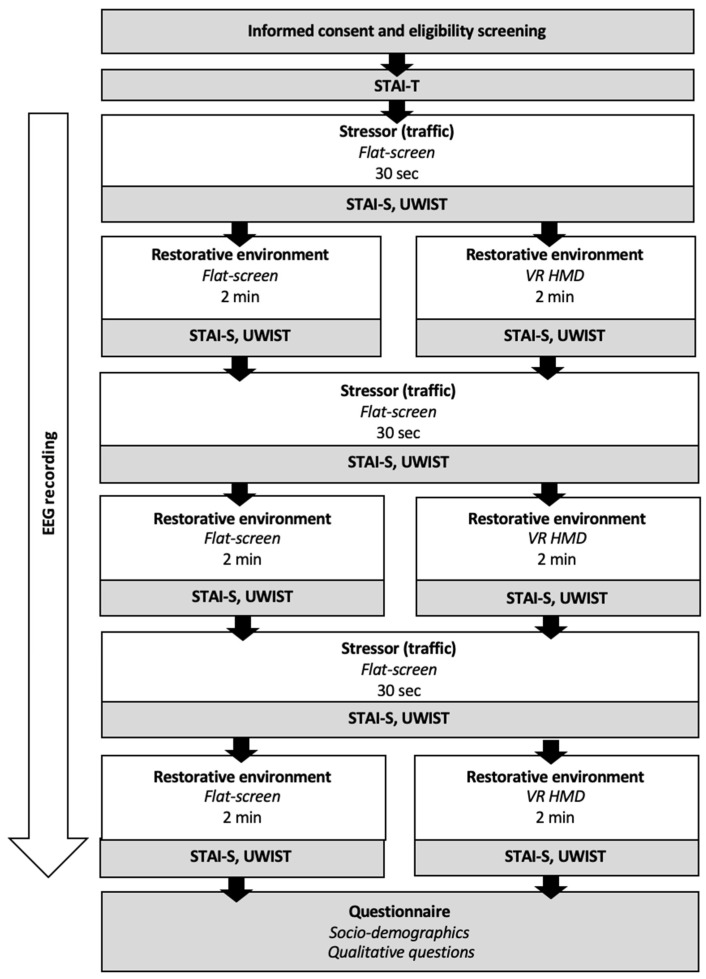
Study design and data collection. Flat-screen = Flat screen computer monitor; VR HMD = Virtual Reality Head-Mounted Display; STAI-T = Subjective Trait Anxiety; STAI-S = Subjective State Anxiety; UWIST = Subjective Stress.

**Figure 2 ijerph-19-09457-f002:**
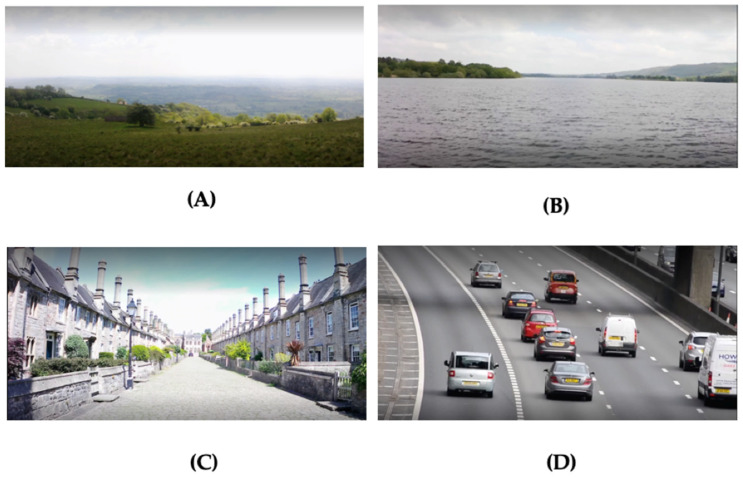
Visual stimuli. (**A**) Green environment; (**B**) Blue environment; (**C**) Historic environment; (**D**) Traffic.

**Figure 3 ijerph-19-09457-f003:**
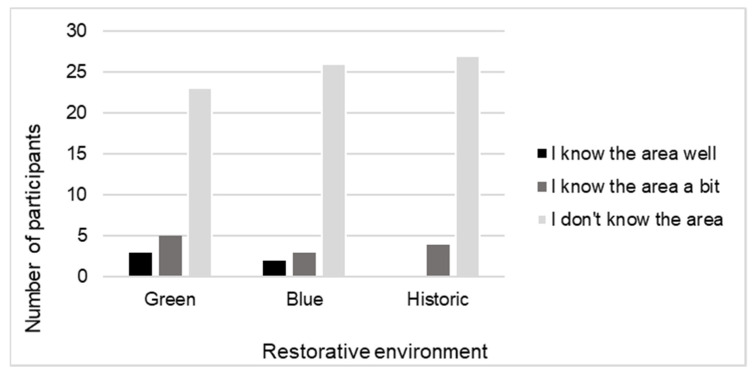
Familiarity with green, blue, and historic environment videos.

**Figure 4 ijerph-19-09457-f004:**
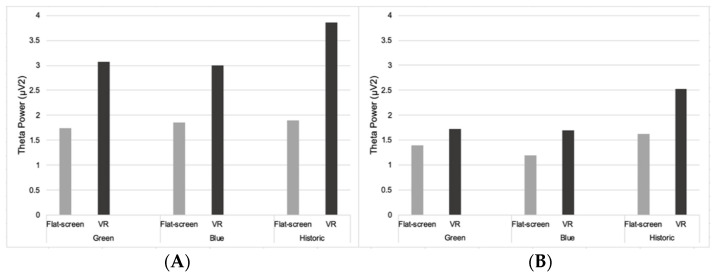
Theta power (µV**²**) in the overall frontal (**A**) and occipital (**B**) regions.

**Figure 5 ijerph-19-09457-f005:**
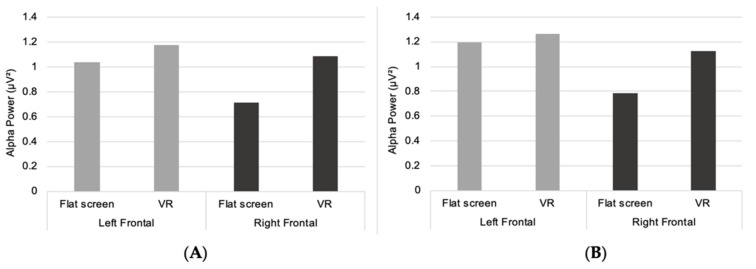
Hemispheric differences in frontal alpha power (µV**²**) for green (**A**) and blue (**B**) environments.

**Table 1 ijerph-19-09457-t001:** Participant demographics per condition.

Demographic		VR	FS
Mean age (yrs)		24.33	24.31
Sex	Female	9	13
	Male	6	3
Ethnicity(*n*)	White-British	8	8
	White-Other	5	3
	Black or Black British—African		3
	Asian—or other Asian background		1
	Mixed—White and Asian	1	
	Not stated	1	

**Table 2 ijerph-19-09457-t002:** Subjective outcome measures (STAI-S and UWIST) after exposure to each environment.

Environment	STAI-S M(SD)	Stress (UWIST) M(SD)
Traffic (1st video)	2.45 (0.26)	2.10 (0.73)
Green	2.25 (0.24)	1.40 (0.64)
Blue	2.28 (0.22)	1.39 (0.69)
Historic	2.19 (0.26)	1.60 (0.61)

**Table 3 ijerph-19-09457-t003:** Linear mixed-effects model for psychological outcomes (STAI and UWIST).

Variables	Reduction of State Anxiety (STAI-S) (β Standard Error)	Reduction of Stress (UWIST) (β Standard Error)
Green	0.02 (0.06)	−0.03 (0.17)
Historic	−0.12 *** (0.06)	0.15 (0.17)
Blue (reference category)		
VR	0.09 (0.10)	0.08 (0.23)
FS (reference)		
Age	−0.002 (0.005)	0.03 * (0.01)
Male	0.15 (0.13)	0.03
Female (reference)		
Order 1	0.19 *** (0.07)	−0.13 (0.17)
Order 2	0.17 *** (0.05)	−0.17 (0.17)
Order 3 (reference)		
Baseline STAI (STAI-T)	−0.04 (0.17)	
Constant	0.14 (0.43)	−1.24 *** (0.39)
Observations	93	93
Number of groups	31	31
LR test vs linear model: chi2(5)	22.53	10.27
Log likelihood	−16.95	−110.03
Wald chi2(7)	22.53	6.13
Prob > chi2	0.001	0.52

*** *p* < 0.01, * *p* < 0.1.

## Data Availability

The data presented in this study are available on request from the corresponding author.

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
