# Peer review of "Exposure to Green, Blue and Historic Environments and Mental Well-Being: A Comparison between Virtual Reality Head-Mounted Display and Flat Screen Exposure"

_ijerph, 2022, doi:10.3390/ijerph19159457_

Round 1

Reviewer 1 Report

The study examines the restorative effects of blue, green, and historic environments. The paper offers some interesting notions while I have some comments to improve the quality.

Abstract: EEG needs explanation.

Abstract needs more numerical/quantitative clue of findings and the importance of the study.

Intro: While blue and green environments somehow sound reasonable to investigate, historic one needs further justifications to included in this triangle.

RQ1 needs explanation in the text as why traffic factors are included here.

Materials and Methods: Were participants selected randomly or?

A control group of non-virtual reality would improve the reliability and the key effects of the study. Is there any chance to include some small sample for this purpose?

How did the study control other features of participants so that their attitudes are not changed by those factors while the experiment/survey/procedure was conducted?

Discussion and particularly Conclusion sections are so weak. Why do we need to use VR tech for the study purpose? What other techniques could be used to improve the validity or greater results?

Reviewer 2 Report

Abstract

The abstract states that the research shows how green, blue and historical environments viewed through flat screens and immersive virtual reality technology can offer quantitative and quantitative advantages for the mental health of the population. Although the concepts may be clear from the context, this reviewer would like to be specified a little more about those green, blue and historical environments, with the aim of giving the reader as much background as possible in an abstract.

Introduction

Line 41: Perhaps this is the space in which the authors can enter to define more clearly what they want to express with green or blue spaces.

Line 122: The research is said to aim to triangulate the results of subjective (psychological) and objective (physiological) measures of well-being and responses to open-ended qualitative questions. It would be very interesting to know previously, in this introductory section, some additional information on the methodology of this triangulation.

Materials and Methods

Lines 136-142: It is explained that a series of subjects have been taken into account, while others have been excluded from the study for presenting a series of diseases or incompatibilities. More information is requested on, for example, why people with skin allergies are chosen to be excluded. The choice of analysis cohorts is key for this type of study, so it is essential to be clear about this type of detail.

Line 144: It is questioned whether a study as ambitious and relevant as the one presented has sufficient methodological robustness after having carried out a study with only 31 participants, of whom 71% are women. A reflection on this sample and its composition is suggested. This reviewer does not consider the study background with the sample analyzed in this research phase to be sufficient.

Round 2

Reviewer 1 Report

Thank you for addressing all my concerns.

Author Response

Thank you for reviewing, we are glad that we have addressed your concerns. 

Reviewer 2 Report

I greatly appreciate the authors for their willingness to make changes that make it possible to improve the proposed text. The sole intention of this revision is to provide the text with the highest possible levels of coherence and clarity.

The authors have carried out a global revision of the text, as suggested in the first round of revisions. After the first change of the authors, the text has improved the readability and clarity of the abstract and the introductory part.

Although the reviewer very significantly appreciates the explanations about the methodological doubts, an inconsistency continues to appear in very important sections of the research. For this reason, it is requested again that the subject of the EEG methodology be reconsidered, which seems to be adopted for this investigation without a clear reconsideration exercise for this investigation.

Author Response

Thank you for reviewing and for your comments, we believe that we have now answered your critical points. We have now provided more explanation for our use of EEG throughout the paper, but especially at the end of the introduction section. We have outlined previous EEG studies which have been conducted before, using different exposure methods (flat screen), and provided more justification for why we chose this measure.